# Systems of Care for Treating Severe Acquired Brain Injury: Comparing the United States to Italy

**DOI:** 10.3390/brainsci15090943

**Published:** 2025-08-29

**Authors:** Nicholas J Cioe, Rita Formisano, Gregory O’Shanick, Juliet Haarbauer-Krupa, Valentina Bandiera, Elisa Berardi, Vincenzo Vinicola, Umberto Bivona

**Affiliations:** 1Rehabilitation Counseling, Assumption University, Worcester, MA 01609, USA; nj.cioe@assumption.edu; 2IRCCS, Fondazione Santa Lucia, Post-Coma Unit, 00179 Rome, Italyu.bivona@hsantalucia.it (U.B.); 3Center for Neurorehabilitation Services, Richmond, VA 23225, USA; dro@cnsva.net; 4Department of Pediatrics, Emory University School of Medicine, Atlanta, GA 30322, USA; jhkrupa@me.com; 5Department of Human Science, LUMSA University, 00193 Rome, Italy; v.bandiera1@lumsa.it

**Keywords:** acquired brain injury (ABI), international comparison, disorders of Consciousness (DOC), systems of care

## Abstract

Acquired Brain Injury (ABI) is now widely regarded as a chronic condition but this change in conceptualization has not yet been realized in the way rehabilitation and care are offered and funded in the United States. Similarly, it is widely accepted that an optimized ABI system includes integration across the phases of care and recovery that considers the bio-psycho-socio-ecological (BPSE) dimensions beyond the injury itself. Despite the importance of BPSE factors informing care, typical post-injury care and management remain focused on acute presentation and the biological nature of the injury and there still exists relevant inter-country differences for disorders of consciousness (DoC) neurorehabilitation after severe ABI. This collaboration with Italian colleagues explores and compares the types and locations of rehabilitative services offered in a Post-Coma Unit of neurorehabilitation center in Italy (namely, Santa Lucia Foundation IRCCS in Rome) and in the United States following a “severe” ABI (sABI). This narrative seeks to describe the degree to which both systems utilize a BPSE informed approach to care.

## 1. Introduction

Acquired Brain Injury (ABI), which had historically been viewed as an injury event with a finite recovery, is now widely regarded by medical professionals, individuals who work with those who have acquired an ABI, families of, and those living with an ABI as a chronic condition [1]. This conceptual shift, initiated by the work of Masel & DeWitt [2], has not yet been realized in the way rehabilitation and care are offered and funded in the United States. Similarly, it is widely accepted that an optimized ABI system includes integration across the phases of care and recovery that considers the bio-psycho-socio-ecological (BPSE) dimensions beyond the injury itself [3]. Despite the importance of BPSE factors informing care, typical post-injury care and management remain focused on acute presentation and the biological nature of the injury [3], and there still exists relevant inter-country differences for disorders of consciousness (DoC) neurorehabilitation after severe ABI [4].

This collaboration with Italian colleagues explores and compares the types and locations of rehabilitative services offered in a Post-Coma Unit of neurorehabilitation center in Italy (namely, Santa Lucia Foundation IRCCS in Rome) and in the United States following a “severe” ABI (sABI). It also seeks to describe the degree to which both systems utilize a BPSE informed approach to care.

The image below, developed and published by the Brain Injury Association of America [5], describes the system of care following ABI in the United States (see Figure 1).

Figure 1 describes the care pathway following an acquired brain injury in the United States.

Comparatively, the image below, developed and published by the Italian Health Ministry [6], describes the Italian system of care for severe ABI (see Figure 2).

Figure 2 describes the care pathway following an acquired brain injury in Italy. The figure was freely adapted from “Guidelines for Assistance to People in a Vegetative State and a Minimally Consciousness State” (https://www.salute.gov.it/imgs/C_17_pubblicazioni_1535_allegato.pdf accessed on 5 April 2025).

## 2. Initial Acute Stages of Care

Upon injury occurrence, the system and processes for evaluating the injury and providing necessary life-saving/stabilizing care (i.e., Emergency Evaluation Emergency Department Intensive Care Unit Specialty NeuroTrauma/Polytrauma) is similar in both the United States and Italy. However, it is the nature, intensity, and duration of treatment occurring in the Specialty NeuroTrauma/Polytrauma stage where care systems in the United States and Italy begin to diverge. This divergence continues in nature, intensity, duration, and availability of care options throughout the post-acute phase of care.

## 3. Specialty NeuroTrauma/Polytrauma

The primary factors determining nature, intensity, and duration of treatment at this stage in the continuum are medical stability, level of consciousness/responsiveness to commands, and time since injury. The goal is to achieve medical stability and increase an individual’s level of consciousness as quickly as possible to facilitate transition to the next stage in the continuum.

### 3.1. United States

Individuals with prolonged Disorders of consciousness (pDoC)—lasting 28 days or longer—who are medically stable should be referred to settings staffed by multidisciplinary rehabilitation teams with specialized training to optimize diagnostic evaluation, prognostication, and subsequent management, including effective medical monitoring and rehabilitative care [7]. Unfortunately, there remains tremendous variability in the ability to predict long-term outcome for those who experience pDoC with individuals in an altered state of consciousness often discharged as soon as medically stable to *sub-acute rehabilitation* facilities, skilled nursing facilities, or home due to funding constraints or poor proximity to specialized DoC programs. Sub-acute rehabilitation settings are for individuals who need less-intensive rehabilitation services for longer periods of time because, though they may still be improving, they are not making rapid functional gains. Length of stay in sub-acute rehab can range from 1–4 weeks. This can vary based on the patient’s condition, services provided, and the facility’s discharge criteria. Patients who experience severe Traumatic Brain Injury (TBI) benefit and have better outcomes from continued rehabilitation [8,9].

### 3.2. Italy

A National Consensus Conference established the transfer criteria from intensive care units (ICU) to rehabilitation facilities [10] for people with severe ABI (sABI). The European Academy of Neurology (EAN) also underlined the importance of a multidisciplinary approach and the diagnostic criteria for individuals with pDoC [11]. The high frequency and severity of comorbidities in patients with sABI and pDoC may interfere with the rehabilitation pathway and may compromise outcome [12,13]. Indeed, as it is well known, recurrent infections, cardiovascular complications, neuro-orthopedic disorders and complicated neurosurgical pathway may slow down and sometimes reduce the efficacy of rehabilitation and the functional recovery. The length of stay in post-acute intensive rehabilitation wards may in fact last several months, sometimes with more significant improvements for those cases with longer time of hospitalization [14].

## 4. Comprehensive Integrated Inpatient Brain Injury Rehabilitation

### 4.1. United States

Individuals who are medically stable, responsive to commands, but not able to safely return home after the acute phase of treatment are appropriate for this level of care. An interdisciplinary team approach is typically used at this stage in the continuum. Insurance is a key issue related to entrance into a rehabilitation program.

Preauthorization by insurance companies is required based on Medicare/Medicaid guidelines (i.e., medical necessity and the ability to tolerate 3 h per day or 15 h per week of therapy). This threshold was often the threshold for health insurance companies as part of the Affordable Care Act (ACA) “essential benefits” but amendments and challenges to the ACA have decreased certainty of coverage. If the individual does not qualify for Medicaid or Medicare, their insurance company does not provide authorization for treatment, or they cannot self-fund this level of care, they are usually discharged to nursing facilities or home. Here they remain until they demonstrate improved functioning to return home (rare) or meet the Medicaid guidelines and secure authorization (sometimes this occurs based on qualifying for Medicaid due to depletion of assets or disability determination).

Unfortunately, individuals often develop worsening health conditions requiring hospital re-admission, which may or may not allow them to be reevaluated for a more intensive therapeutic setting. In the US, insurance status frequently dictates one’s ability to obtain both necessary medical treatments and preventive care. Disparities in insurance coverage are prevalent, with minority populations facing greater obstacles in securing health insurance compared to non-Hispanic Whites, consequently diminishing their likelihood of utilizing healthcare services. Research indicates that uninsured individuals are at a disadvantage in terms of regular healthcare provision, resulting in heightened susceptibility to adverse health outcomes [15,16]. Moreover, uninsured individuals are less likely to have regular care and those who experience traumatic brain injury or strokes are also less likely to access rehabilitation care thereby placing them at risk for poorer health outcomes [17,18].

### 4.2. Italy

Post-acute intensive neurorehabilitation recognizes different codes for post-comatose patients. In particular, “code 75” refers to intensive neurorehabilitation of patients with sABI, including post-comatose patients. This code indicates a specialized and intensive level of rehabilitation, aimed at patients with significant disabilities following coma or other brain damage, which includes not only neuromotor rehabilitation, but also a multidisciplinary approach, such as multisensory stimulation, for DoC, neuropsychological rehabilitation, swallowing and respiratory training, and management of withdrawal from tracheal tube and from enteral nutrition. An alternative code, code 56, is assigned to patients with less severe needs but who still require 24-h medical and nursing supervision for a period of intensive medical, nursing and rehabilitation assistance.

It is fundamental to the Italian system of care that an inter-professional method of the multidisciplinary approach is utilized at this stage in the continuum. This approach implies that each professional of the rehabilitation team (e.g., physician, neuropsychologist, speech/physio/occupational therapists, nurse, etc.) work in mutual and constant cooperation. This system stresses the importance of collaborating with the patient and their informal caregiver throughout the care management and rehabilitation process. This in-rehabilitation pathway implies a “patient-tailored” approach, with a mean length of stay of six months (and, in some cases even one year, according to the severity of the post-sABI consequences), with longer in-rehabilitation stays (162 days on average) associated with higher improvement rates [14].

The Post-Coma Unit system described above implies the full involvement of at least one of the patient’s informal caregivers. An essential component is the caregiver receives psycho-educational support from a clinical psychologist, aimed at providing individualized information to best support their loved one (thus, extending the rehabilitation process beyond each rehabilitation setting). Informal caregivers also received psychological support aimed at treating the severe psycho-emotional caregiver burden [19,20,21]. Indeed, it has been demonstrated that the caregivers’ psychological well-being was associated with the features of caregiving, the subjective approach to neuro-rehabilitation, and the functional recovery of their loved ones.

Interestingly, better caregivers’ approach to neuro-rehabilitation was also associated with an overall positive impact on caregiving in neuro-rehabilitation and to a better functional outcome of the patients [22,23]. The study by De Luca and colleagues [23] confirmed the importance of caregiver *physical* presence by comparing the effect of informal caregiver’s *physical* presence on their loved one compared to those whose caregivers were only able to assist *remotely* (through videocalls). Study findings demonstrated that, although both groups of patients improved after the treatment, the improvement was consistently greater in the group assisted by caregivers who were physically present versus remote.

## 5. Post-Acute Residential Transitional Rehabilitation

### 5.1. United States

This level of care is critical to maximizing functioning and independence with the expectation of individuals participating in at least six hours of therapy per day. This type of specialized ABI rehabilitation is considered the gold standard but unfortunately fewer than 25% of individuals who survive moderate, severe, or penetrating TBI receive this level of care [24].

Individuals engaging in this type of rehabilitation usually receive specialized individual and group therapies in cognitive rehabilitation, speech-language, physical, occupational, vocational, recreational, behavior, and psychotherapy. The interdisciplinary team is led by a case manager who coordinates with specialized medical providers (e.g., physiatry, psychiatry, ophthalmology, orthopedic) as needed, maintains communication with the funder, and assures effective communication among clinical team members, organizational staff, and the family/supports of the individual with an ABI. Person-centered models of care ensure that the individual with an ABI is included in goal setting and decision making. Of primary importance is functioning in the necessary life domains to include activities of daily living (ADLs) (e.g., bathing, dressing, eating), instrumental activities of daily living (iADLs) (e.g., meal preparation, grocery shopping, scheduling, transportation), and vocational (e.g., school, work).

The nature of this level of care is *transitional* and is meant to assist individuals transitioning from more intensive acute or hospital-based rehabilitation settings to less intensive home and community-based supportive or outpatient services. The length of admission is less influenced by clinical judgement and more influenced by funding coverage. Initial evaluation periods are usually less than two weeks, and lengths of stay often involve multiday, weekly, or monthly extensions. The funder reporting requirements are often an unnecessary factor influencing the intensity of treatment instead of matching the intensity and duration of treatment to what is tolerable or clinically recommended. Length of stay can range from 10 days to 12+ months, though nowadays it is rare for stays to exceed 9 months. Individuals discharge to independent living, home with family and supportive day-treatment or outpatient services, supportive living group home or individual apartment settings, or, if they have not progressed or have regressed (back to a sub-acute setting). If they have care needs that are not supported by insurance or private funds that cannot be met in a home environment, they often are discharged to nursing home settings.

### 5.2. Italy

Typically in the Italian system, specialized holistic rehabilitation is not provided in community-based settings. Rather, intensive interdisciplinary rehabilitation is provided in a rehabilitation hospital setting at the Day Hospital and outpatient rehabilitation pathway. The need for family education and support is essential because they absorb the survivor’s care needs when independent functioning is not possible. As such, rehabilitation services and length of stay are influenced by discharge placement and available support. Recovery potential and improvement toward independent living, under the guidance of clinical judgment, dictate the rehabilitation pathway. Provision of care through outpatient remains funded by the public system and private insurance or private pay are rarely involved in long-term rehabilitation and social reintegration programs.

## 6. Access Barriers

### 6.1. United States

The most significant determinant of whether a person with ABI receives rehabilitative services is funding. Individuals with extreme wealth or those who already qualify for Medicare or Medicaid usually have the fewest barriers to access. Most individuals in the United States have private health insurance (usually provided through their employer). The coverage provided by these plans varies greatly and the process of navigating these processes is often daunting and confusing.

Providers at the previous stage in the continuum (Acute care) encounter multiple barriers in addition to the limited knowledge and awareness they may have to assure optimal referral. Barriers may also include: pressure to move patients out of beds is in contrast with the multiday preauthorization or preadmission screening requirements, expectations from post-acute providers that the individual will be able to discharge home upon completion of post-acute care, reluctance of families to have their loved one sent to programs that may be far from their home or in a different state, and social determinants of health influence decision making and trust [25].

### 6.2. Italy

After an intensive neurorehabilitation pathway, lasting between six months and one year, for persons with pDoC, long-term nursing homes are chosen by a case manager, with different levels of assistance (e.g., tracheal tube, enteral nutrition, high occurrence of comorbidities). These placements are partially supported by the public health system according to the intensity level of assistance and the family income.

For patients with recovery of consciousness and/or neuropsychological improvement, home reintegration and neurorehabilitation projects in Day Hospital (DH) regimen are the preferred choice, especially for parents of young patients. This new setting, always supported by the Italian public system, may last from three to six months.

## 7. Supportive Community-Based Services

### 7.1. United States

Almost every community in the country has access to physical therapy and psychotherapy (though the presence of the therapy does not necessitate specialized knowledge to support individuals with ABI). Only moderately sized towns and cities close to major cities or college/universities are likely to have specialized therapies (e.g., cognitive, speech language, occupational, applied behavior analysis), or specialized medical providers to meet the needs of individuals with ABI. Availability and access to day treatment or other specialized care options for individuals with ABI are even more sparsely available. However, there are some states with excellent care options throughout the continuum of care (e.g., Colorado, Michigan, Massachusetts, Florida, Texas). These states often have ABI waiver funding streams to underwrite the cost of care for individuals in need and have health care regulations that provide a roadmap for ABI rehabilitation companies to implement profitable business models. Unfortunately, the presence or absence of these services is determined by proximity.

### 7.2. Italy

Public health supports not only DH but also outpatient rehabilitation services, which may offer several months of further rehabilitation (e.g., neuropsychological training, speech/physio/occupational therapies, hydro-kinesitherapy, swallowing and phoniatric treatment, orthoptic treatment and psychoeducational support to the patients’ informal caregivers). More specific programs (e.g., aiming at the recovery of driving abilities) are also available. The Italian system does have guidelines for return to school processes but vocational rehabilitation and return to work processes are still lacking (though Italian laws related to work reintegration for people with disabilities exist).

## 8. Special Conditions and Circumstances

### 8.1. United States

The for-profit health care system that drives access to care in the United States is not necessarily the system faced by all individuals who may acquire an ABI. Special populations (e.g., service members and Veterans who have health coverage through the military or Veterans Administration) operate within a different system of care. While the evaluation process may differ, the guaranteed coverage (like the Italian healthcare system) eliminates access barriers and allows for seamless transition throughout the continuum of care based on clinical recommendation in consultation with the individual and their support system. There remain proximity limitations depending on where the individual lives, were when they were injured, rehabilitation needs, etc., but the influence of funding is removed from the equation.

Similarly, individuals who acquire ABI while at work often do not face the same obstacles to the care continuum as the general population. The United States’ workers compensation insurance system oversees a worker’s injury, providing 500 weeks of base salary of the injured party, and “owns” that injury until the person returns to their prior level of employment. Should the injured worker at the end of 500 weeks be unable to return to work due to permanent brain injury, they are entitled to a lifetime salary and medical benefits from the employer of record. An injured employee’s inability to return to their prior level of employment is considered some form of permanent disability, which may result in lifetime wage supplements, care arrangements, or settlements with the insurer. This historically has incentivized Worker’s Compensation companies to provide optimal care throughout the continuum to maximize recovery and decrease long-term liability; however, a trend towards more aggressive review and denial of benefit claims has evolved over the past decade.

### 8.2. Italy

In Italy, only a low percentage of people with sABI are supported for rehabilitation by private health insurance. Instead, the public health system sustains the whole pathway of rehabilitation and long-term care, especially for people with pDoC. If patients do have private health insurance, they can enjoy further advantages (e.g., outpatient neurological and neuropsychological consultation and evaluation, psychological support to their caregivers or the patients themselves), even after the end of the rehabilitation process, which is when many ancillary services funded by the public system would end. Also, work connected injuries present with some advantages in Italy, thanks to the support of the National Institute for the Insurance against Work Injuries (INAIL), either in terms of financial compensation or rehabilitation benefits.

## 9. Summary & Opportunities

The present comparison between the USA and Italian neurorehabilitation system evidenced that although patients are similarly assisted in the acute stage, many differences emerged in the post-acute phase, especially in terms of length of stay in the rehabilitation setting, access to services associated with funding, prioritization of informal caregiver involvement, and availability of home and community-based services (the latter two represent important components of the BPSE approach). Table 1 highlights the main differences between the American and Italian neurorehabilitation systems of care.

Advanced intensive care and early neurosurgical approach in the acute phase progressively increased the survival rate of people with sABI and their neurorehabilitation needs. Moreover, in recent years, a significant reduction in the duration of acute phase hospital stays, per episode of care, resulted in longer stays in post-acute care facilities [26], since even more patients are discharged “quicker and sicker” from the acute phase [14].

The duration of some United States rehabilitation programs lasting only a few weeks may often be insufficient because of the high occurrence of recurrent comorbidities [12,13,27], which sometimes may also delay the transfer from the ICU to rehabilitation facilities in the acute phase. As an example of comparison, in calendar year 2024, post-Coma Unit of Santa Lucia Foundation patients length of stay were 125 days for inpatient (M = 85, F = 57; with a mean age of 55 years and a mean educational level of 12 years) and 149 days for patients in the day hospital (DH) (M = 81, F = 38; with a mean age of 49 years and a mean educational level was of 13 years). These lengths of stay correlated with significant improvements in mean Disability Rating Scale (DRS) [28] scores and improvements in function (see Table 2).

Thirty years of rehabilitation practice in the Post-Coma Unit of the Santa Lucia Foundation demonstrated the need for and the effectiveness of inter-professional and patient-centered approaches, which typically are a recognized and practiced component in the United States. However, the prioritization of including informal caregivers in the rehabilitation process of patients with sABI to positively contribute to the efficacy of the rehabilitation program is more present in the Italian system. Unfortunately, proposed changes to accreditation criteria for high specialty rehabilitation wards in Italy are putting the mandatory inclusion of education and involvement of informal caregivers at risk. This change jeopardizes this important aspect of the Italian rehabilitation system, especially given the familial/societal cultural differences (collective vs. individualistic) between Italy and the United States.

The United States system, regarding home and community-based services, exceed the availability of similar services in Italy because most of those services are provided during longer in-patient stays or as part of day-hospital programs in Italy. Like the US, proximity plays a significant role in participation, whether in a hospital-based system (Italy) or a home and community-based system (USA).

Perhaps the most significant difference between the Italian and United States systems of care are associated with access to services based on funding. The guaranteed access to care throughout the continuum funded by the Italian public health care system vastly differs from the complex and hard to navigate private health insurance system in the United States. Except for the differences in access associated with military or work-connected injuries, most individuals experiencing sABI in the United States face significant barriers to accessing necessary care along the continuum without interruptions. The consequences of these access issues and interruptions in care on long-term outcomes contributes to the massive societal cost of brain injury in the United States.

The researchers hope this level of international comparison is only the beginning. Further exploration of these and other international systems of care and data could offer extensive insights into system weaknesses and areas for improvement of services. For example, neurotrauma services positively influence the management of the TBI population in different clinical settings [29]. Further research and the collaboration of global clinicians to develop international guidelines, especially ones also applicable to low-to-middle income countries, are warranted [29]. In this perspective, Ferioli et al. [30], by means of an international survey, created a preliminary international map of centers specialized in pDoC (https://www.google.com/maps/d/viewer?ll=41.9195397130714%2C12.651486480534881&z=10&mid=1lpRu92tElS-zikW6mOoTO-qZl9G-y70 accessed on 22 August 2025). Future research will aim to identify additional relevant care centers, broaden geographical representation, and strengthen collaboration to enhance accessibility and improve outcomes for patients with pDoC.

Our intention is not to criticize or compliment either model. Instead, the hope is that those involved in policy development can consider alternative systems of care as they seek to develop and improve their systems.

## 10. Limitations

Although this study seeks to compare systems of care following sABI in two different countries, it does not adequately address the intra-country diversity in systems and experiences. The United States system description is a generalization with significant variability at the state and community level. The same is true in Italy with significant variability amongst the regions with regard to length of stay and complexity of the rehabilitation process. Also, the impact of cultural norms (historical and emerging) must be considered when exploring systems of care. 

## Figures and Tables

**Figure 1 brainsci-15-00943-f001:**
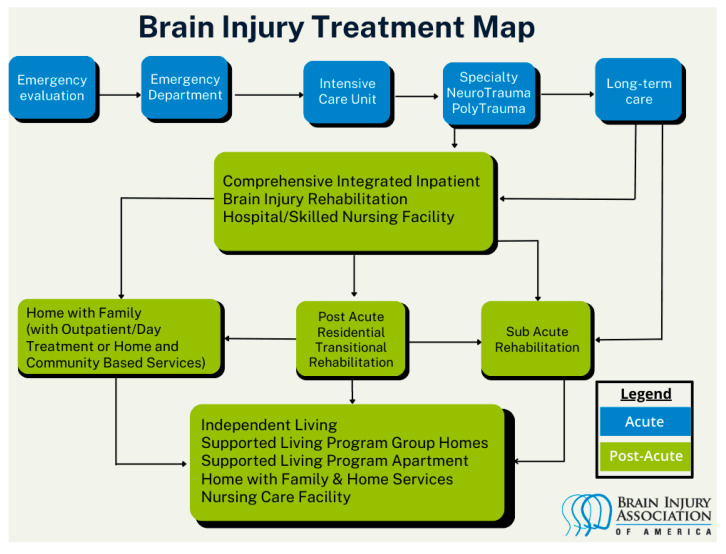
United States Brain Injury Treatment Map.

**Figure 2 brainsci-15-00943-f002:**
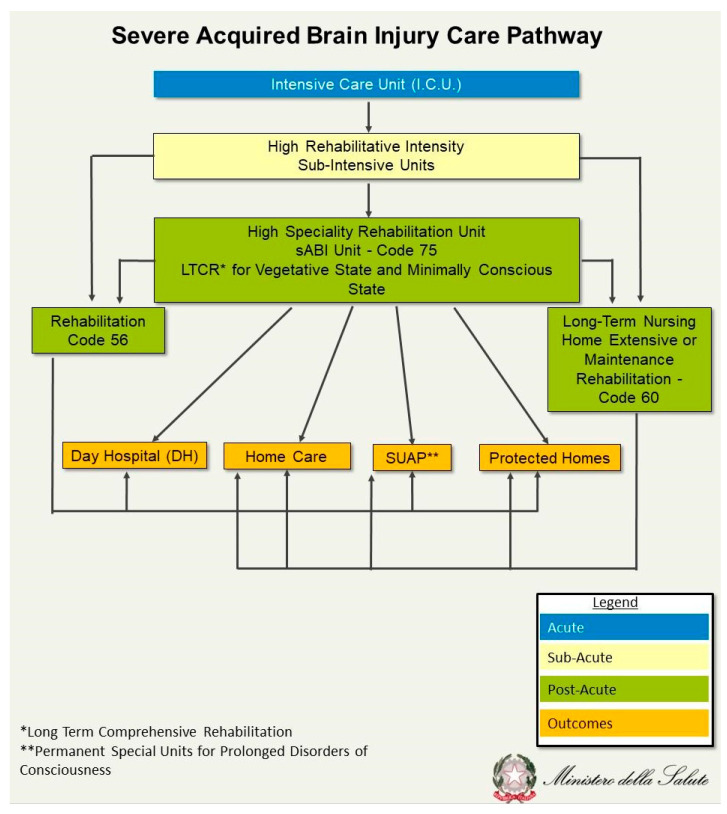
Italian Brain Injury Care Pathway.

**Table 1 brainsci-15-00943-t001:** Main Differences between American and Italian post-acute systems of care. Following severe acquired brain injury (sABI).

	*United States*	*Italy*
**Primary Funding Source**	Private Health Insurance	Public
**Length of Stay (avg)**	<21 days	4+ months
**Caregiver Involvement**	Case Specific	Essential
**Access to initial post-acute care**	Not guaranteed; if authorized, closely scrutinized	Automatically provided based on medical/clinical recommendation
**Access to Community Based Services**	Robust	Minimal

**Table 2 brainsci-15-00943-t002:** Level of disability (according to the DRS score) changes in rehabilitation in Santa Lucia Foundation in 2024.

	In-Patient Level of Disability	Day Hospital Level of Disability
Admission *(disability)*	16 *(severe)*	7 *(moderate–severe)*
Discharge *(disability)*	10 *(moderate–severe)*	5 *(moderate)*

## Data Availability

The data presented in this study are available on request from the corresponding author due to it being (proprietary aggregate outcome data from Fondazione Santa Lucia).

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
