# Peer review of "Systems of Care for Treating Severe Acquired Brain Injury: Comparing the United States to Italy"

_brainsci, 2025, doi:10.3390/brainsci15090943_

Round 1
Reviewer 1 Report
Comments and Suggestions for Authors
The manuscript titled "Systems of Care for Treating Severe Acquired Brain Injury: Comparing the United States to Italy" presents a comprehensive and well-articulated comparative analysis of the healthcare systems in the United States and Italy regarding the management of severe acquired brain injury (ABI). The paper offers a highly informative perspective on the structural, organizational, and operational differences between the two countries, with specific attention to the nature, intensity, duration, and accessibility of post-acute care services.
The narrative is clearly structured. However, I would like to offer a few suggestions to enhance the clarity and utility of the manuscript:
- While the narrative provides rich detail, the addition of a summarizing table that consolidates the most important differences between the Italian and U.S. systems would significantly enhance the readability and comparative value of the paper.
- The section on “Post-Acute Residential Transitional Rehabilitation” currently provides information exclusively on U.S. practices. For completeness and balance, the authors should include corresponding details about the similar services in Italy.
- The table presented in the “Summary & Opportunities” section lacks a formal title or label. For clarity and proper referencing, the authors should provide a descriptive caption and table number.
Overall, the manuscript makes a valuable contribution to the field of neurorehabilitation and cross-national healthcare systems research. Addressing the above points would further improve the paper’s clarity and accessibility.
Reviewer 2 Report
Comments and Suggestions for Authors
Dear Editor and authors:
I have had the opportunity to review the manuscript entitled “Systems of Care for Treating Severe Acquired Brain Injury: Comparing the United States to Italy”, submitted as a perspective article. I appreciate the chance to contribute to the editorial process.
First, I would like to acknowledge the relevance of the topic addressed. The comparative focus on the United States and Italy in the care of severe acquired brain injury is both timely and pertinent, given the growing need for integrated models of care that incorporate the bio-psycho-socio-ecological (BPSE) framework. Moreover, the international collaboration adds significant value to the manuscript.
Below I provide my comments regarding the article.
-Including images or comparative diagrams of the care systems (such as those from the Brain Injury Association of America and the Italian Ministry of Health) adds value and clarity to the manuscript. Just be sure to properly cite these sources and obtain permissions if necessarY.
-Including full URLs directly within the main text is generally not standard practice in formal academic manuscripts. Typically, it is preferred to cite the source by the organization name and date within the text, and then provide the full URL in the reference list or as a footnote if necessary. This approach keeps the manuscript visually clean and aligns with most journal guidelines. For example, instead of inserting the full URL in the introduction, you might cite it as (Brain Injury Association of America, n.d.) and include the full link in the references section (on page 2 of the manuscript. Since there are no line numbers available for referencing).
-In the sentence ‘The image below, developed and published by the Brain Injury Association of America (https://biausa.org/brain-injury/about-brain-injury/treatment)’, the word ‘America’ is broken at the end of the line, which affects readability.
-In the section comparing post-acute care between Italy and the United States, it is noted that the Italian case explicitly mentions a specific physical location (Modena, 2000) as a reference for the national consensus on criteria for transfer from ICU to rehabilitation. However, in the United States section, references are made to national guidelines and studies without citing a specific location or similar event.
It would be advisable to clarify this difference in the text to avoid potential confusion. For example, it could be explained that the reference to Modena corresponds to a formal, localized event, whereas in the United States the regulations or practice guidelines are of a national and decentralized nature, which justifies the absence of a precise geographic reference. This would help improve the reader’s understanding of the structural and contextual differences between the two healthcare systems.
-Some sections are quite lengthy, which may make them difficult to follow. Consider summarizing or breaking them into shorter paragraphs to improve clarity and readability.
- I would recommend including one or more comparative tables to enhance clarity and readability. For instance, a table summarizing the main differences between the U.S. and Italian neurorehabilitation systems (e.g., funding sources, length of stay, caregiver involvement, access to community-based services, and long-term outcomes) would allow readers to grasp the contrasts at a glance. Presenting the reported data (such as average length of stay, Disability Rating Scale improvements, and coverage mechanisms) in a structured format would also reduce the density of the text and improve overall comprehension.
-In the Summary & Opportunities section, you might consider explicitly highlighting potential policy implications arising from the comparison. For instance, what lessons could be transferred between the U.S. and Italian systems to reduce costs and improve outcomes? As examples, the Italian model of systematically including informal caregivers could inform U.S. practices, while the U.S. emphasis on home and community-based services may offer strategies adaptable to the Italian context.
REFERENCES
The reference format currently used in the manuscript follows the APA author-date style, which is not consistent with MDPI guidelines that require citations in a numerical style enclosed in brackets [ ].
AUTHOR CONTRIBUTIONS SECTION
Typically, in the Author Contributions section, it is standard practice to use authors’ initials rather than their full names for clarity and brevity. For example, instead of writing ‘Nicholas Cioe and Umberto Bivona’, it would be preferable to use ‘N.C. and U.B.’.”
Reviewer 3 Report
Comments and Suggestions for Authors
I actually enjoyed reading this manuscript with compares the offer of care in acute neurorehab setting for severe acquired brain injury patients in the United States and Italy.
The authors provide a good description of the similarities and differences between the two countries when it comes to triage, assess and accept referrals for those patients and the barriers to provide excellence of clinical care including availability of funding.
The quality of the images is good and helps identifying the key aspects of the clinical pathways in the two countries. The methodology for comparison is less clear and I would suggest the authors improve this by clarifying that this study consists in a narrative review aimed at identifying opportunities for improving existing pathways (rather than a pre-defined clinical investigation).
Furthermore, given the neurotrauma/polytrauma aspect of many ABI I would suggest including the work from Dasic et al in your reference list and stating that enhancing the pace of transition from acute care to neuro-rehab centers has been identified in the matrix of currently unmet needs of neurotrauma services in major trauma centers:
Ref: Dasic D, Morgan L, Panezai A, Syrmos N, Ligarotti GKI, Zaed I, et al. A scoping review on the challenges, improvement programs, and relevant output metrics for neurotrauma services in major trauma centers. Surg Neurol Int. 2022;13:171. doi: 10.25259/SNI_203_2022.
Many thanks again for the opportunity to review this study and I look forward to receiving your revised manuscript soon.
Round 2
Reviewer 3 Report
Comments and Suggestions for Authors
The authors have revised well this narrative review